# Antioxidant Activities and Cytotoxicity of the Regulated Calcium Oxalate Crystals on HK-2 Cells of Polysaccharides from *Gracilaria lemaneiformis* with Different Molecular Weights

**DOI:** 10.3390/foods12051031

**Published:** 2023-02-28

**Authors:** Jing-Hong Liu, Yu-Yun Zheng, Jian-Ming Ouyang

**Affiliations:** Institute of Biomineralization and Lithiasis Research, College of Chemistry and Materials Science, Jinan University, Guangzhou 510632, China

**Keywords:** *Gracilaria lemaneiformis* polysaccharide, antioxidant activity, cytotoxicity, calcium oxalate, molecular weight

## Abstract

The antioxidant activities of seven degraded products (GLPs) with different molecular weights (*M*_w_) of polysaccharides from *Gracilaria lemaneiformis* were compared. The *M*_w_ of GLP1–GLP7 were 106, 49.6, 10.5, 6.14, 5.06, 3.71 and 2.42 kDa, respectively. The results show that GLP2 with *M*_w_ = 49.6 kDa had the strongest scavenging capacity for hydroxyl radical, DPPH radical, ABTS radical and reducing power. When *M*_w_ < 49.6 kDa, the antioxidant activity of GLPs increased with the increase in *M*_w_, but when *M*_w_ increased to 106 kDa, their antioxidant activity decreased. However, the ability of GLPs to chelate Fe^2+^ ions increased with the decrease in polysaccharide *M*_w_, which was attributed to the fact that the polysaccharide active groups (–OSO_3_^–^ and –COOH) were easier to expose, and the steric hindrance was smaller when GLPs chelated with Fe^2+^. The effects of GLP1, GLP3, GLP5 and GLP7 on the crystal growth of calcium oxalate (CaOx) were studied using XRD, FT-IR, Zeta potential and thermogravimetric analysis. Four kinds of GLPs could inhibit the growth of calcium oxalate monohydrate (COM) and induce the formation of calcium oxalate dihydrate (COD) in varying degrees. With the decrease in *M*_w_ of GLPs, the percentage of COD increased. GLPs increased the absolute value of the Zeta potential on the crystal surface and reduced the aggregation between crystals. Cell experiments showed that the toxicity of CaOx crystal regulated by GLPs to HK-2 cells was reduced, and the cytotoxicity of CaOx crystal regulated by GLP7 with the smallest *M*_w_ was the smallest, which was consistent with the highest SOD activity, the lowest ROS and MDA levels, the lowest OPN expression level and the lowest cell necrosis rate. These results suggest that GLPs, especially GLP7, may be a potential drug for the prevention and treatment of kidney stones.

## 1. Introduction

Kidney stone is a disease that seriously endangers people’s health. The prevalence of kidney stones in adults in China is 5.8% (6.5% in men and 5.1% in women) [1], and about 9% of people around the world are affected by kidney stones [2]. Kidney stones have not only a high prevalence, but also a high recurrence rate. Kidney stones can be roughly divided into six types by composition: calcium oxalate (CaOx), calcium phosphate, uric acid (urate), magnesium ammonium phosphate, cystine and purine stones, among which CaOx (including calcium oxalate monohydrate (COM) and calcium oxalate dihydrate (COD)) accounts for about 80–84% [3]. No matter what type of kidney stones, their formation is a complex multi-step process, including urine supersaturation, crystal nucleation, growth and aggregation [4].

COM crystals are more common than COD crystals in clinical stones, occurring about twice as often as COD crystals. Generally, COM crystals are mostlylarge aggregates with sharp edges and corners, which are highly toxic to renal epithelial cells, while COD crystals are numerous small-sized round blunt particles with fewer aggregates that are relatively less toxic to cells. Our previous study [5] showed that COM crystals exhibited higher cytotoxicity than COD crystals at the same size and crystal concentration, resulting in more LDH release, higher ROS level and lower mitochondrial membrane potential.

Gracilariales is a kind of macroalgae with more than 300 species, of which 160 species have been accepted by taxonomy [6]. *Gracilaria lemaneiformis* grows in tropical and subtropical waters around the world. It has strong adaptability to the environment, fast growth and rich resources. *Gracilaria lemaneiformis* polysaccharide (GLP) is the main component of *Gracilaria lemaneiformis*. Its structure is generally composed of alternating β-1,3 and α-1,4 D and L galactose residues [7]. GLP is an acidic polysaccharide rich in sulfate, with antioxidant, antiviral, anticoagulant and anti-tumor effects [8,9,10].

Natural polysaccharides are usually heavy in molecular weight (*M*_w_). It is difficult for cells to enter through the body barrier, affecting their biological activity [11,12,13]. Degradation of polysaccharide to appropriate *M*_w_ can optimize their biological activity. Liu et al. [11] obtained two products, GLP_L_1 and GLP_L_2, with *M*_w_ of 5.2 kDa and 15.4 kDa by degrading original *Ganoderma lucidum* polysaccharides (GLPP). At a concentration of 10 mg/mL, the antioxidant activity of GLP_L_1 was significantly higher than that of GLP_L_2, and the scavenging rates of GLP_L_1 and GLP_L_2 on hydroxyl radicals (·OH) were 78.3% and 53.6%, respectively. Sun et al. [12] obtained three polysaccharides, EPS-1, EPS-2 and EPS-3, with *M*_w_ of 256.2 kDa, 60.7 kDa and 6.55 kDa, by degrading *P. purpurea* polysaccharides with *M*_w_ of 2918 kDa using the microwave method. When the polysaccharide concentration was 2.0 mg/mL, the DPPH free radical scavenging rate was EPS-3 (83.8%) > EPS-2 (78.3%) > EPS-1 (38.7%). That is, the polysaccharide with low *M*_w_ has higher antioxidant activity. Liao et al. [7] studied the hypoglycemic and antioxidant effects of polysaccharides extracted from *Gracilaria lemaneiformis* (GLP, 121.89 kDa) and their degradation products (GLP1, 57.02 kDa; GLP2, 14.29 kDa) in alloxan-induced diabetic mice. At the same dose, GLP1 showed better efficacy than GLP and GLP2. This suggests that it is not that the lower the *M*_w_ of a polysaccharide, the better its bioactivity.

In the previous study [14], we degraded the original *Gracilaria lemaneiformis* polysaccharides (GLP0) with *M*_w_ = 622 kDa and obtained seven degraded polysaccharides: GLP1, GLP2, GLP3, GLP4, GLP5, GLP6 and GLP7, with *M*_w_ of 106, 49.6, 10.5, 6.14, 5.06, 3.71 and 2.42 kDa, respectively. We also characterized their structures. Before and after degradation, the content of the characteristic group –OSO_3_^–^ in the polysaccharides did not change much (13.26% ± 0.20%). GLPs have the ability to repair human renal proximal tubular epithelial cells (HK-2) damaged by oxalic acid, and the repair ability is closely related to polysaccharide *M*_w_. GLP2, with *M*_w_ = 49.6 kDa, had the strongest repair ability, and too large or too small *M*_w_ led to a decrease in polysaccharide repair ability.

This paper further studied the differences in the antioxidant activity of these polysaccharides, as well as the regulation effects of these polysaccharides on the growth of CaOx crystals and the cytotoxicity difference of the regulated CaOx crystals, hoping to provide enlightenment as to finding the best activity of polysaccharides for the prevention and treatment of kidney stones.

## 2. Experiments

### 2.1. Reagents and Instruments

*Gracilaria lemaneiformis* polysaccharide (GLP0) was provided by Beijing Newprobe Bioscience & Technology Co., Ltd. (Beijing, China). 1,1-diphenyl-2-trinitrophenylhydrazine (DPPH), 2,2′-biazobis (3-ethylbenzothiazolin-6-sulfonic acid) diammonium salt (ABTS), phenazine (Ferrozine), ascorbic acid (Vc), ethylenediaminetetraacetic acid (EDTA), ferrous sulfate (FeSO_4_), phosphate buffer solution (PBS), potassium persulfate (K_2_S_2_O_8_), potassium ferricyanide (K_3_[Fe(CN)_6_]) and o-diazepine were purchased from Shanghai Aladdin Biochemical Technology Co., Ltd. (Shanghai, China). Analytically pure conventional reagents such as trichloroacetic acid (CCl_3_COOH), 30% hydrogen peroxide (H_2_O_2_), sodium oxalate (Na_2_Ox) and calcium chloride (CaCl_2_) were purchased from Guangzhou Chemical Reagent Factory (Guangzhou, China). The experimental water was secondary distilled water.

Human proximal tubular epithelial cells (HK-2) were purchased from Shanghai Cell Bank, Chinese Academy of Sciences. The total superoxide dismutase (SOD) assay kit and malondialdehyde (MDA) assay kit were purchased from Nanjing Jiancheng Bioengineering Institute (Nanjing, China). OPN antibody and goat anti-mouse IgG-FITC were purchased from Wuhan Boster Biological Technology Co., Ltd. (Wuhan, China). Cell Counting Kit-8 (CCK-8 kit), reactive oxygen species detection kit and 4′,6-diamidine-2-phenylindole (DAPI) dye were purchased from Shanghai Beyotime Bio-Tech Co., Ltd. (Shanghai, China). Hoechst 33342/PI double staining kit was purchased from Beijing Solarbio Technology Co., Ltd. (Beijing, China). Penicillin–streptomycin solution (100×), trypsin, fetal bovine serum (FBS) and DMEM/F12 medium (Gibco, Grand Island, NY, USA). Cell culture plates and confocal dishes were purchased from Wuxi NEST Biotechnology Co., Ltd. (Wuxi, China).

The Instruments used in this paper were as following: UV-Vis-NIR spectrophotometer (Cary 5000, Agilent, Santa Clara, CA, USA), X-ray powder diffractometer (D/MAX2400, Rigaku, Tokyo, Japan), X-L environmental scanning electron microscope (ESEM, Philips, Eindhoven, The Netherlands), nanoparticle size-Zeta potential analyzer (Zatasizer 300HS, Malvern, Worcestershire, UK), thermogravimetric analyzer (TGA/DSC 3+, Mettler Toledo, Zurich, Switzerland), inverted fluorescence microscope (DMRA2, Leica, Wetzlar, Germany), enzymometer (Safire2, Tecan, Männedorf, Switzerland), laser scanning confocal microscope (LSM 800, Zeiss, Jena, Germany).

### 2.2. Degradation of GLP0

According to reference [14], GLP0 with *M*_w_ = 622 kDa was degraded by 30% H_2_O_2_ to obtain seven degraded polysaccharides with *M*_w_ of 106, 49.6, 10.5, 6.14, 5.06, 3.71 and 2.42 kDa, respectively, named GLP1, GLP2, GLP3, GLP4, GLP5, GLP6 and GLP7. Their structures were characterized by FT-IR, ^1^H NMR and ^13^C NMR.

### 2.3. Antioxidant Activity of GLPs

#### 2.3.1. Scavenging Hydroxyl Radicals (·OH) of GLPs

The H_2_O_2_/Fe^2+^ system was used to test the scavenging ability of polysaccharides on ·OH in vitro [15]. That is, FeSO_4_ solution (2.5 mmol/L, 1 mL) and phenanthroline solution (2.5 mmol/L, 1 mL) were added into test tubes, and then PBS with pH = 7.4 (20 mmol/L, 1 mL), H_2_O_2_ (20 mmol/L, 1 mL) and polysaccharide solution with different concentrations (0.15–3 mg/mL, 1 mL) were added successively. After mixing evenly, the temperature was kept at 37 °C for 90 min, and then the absorbance was measured at 536 nm by a UV-Vis-NIR spectrophotometer, which was repeated three times to obtain the average absorbance (A_3_). In the undamaged group, the volume was supplemented with distilled water without sugar solution and hydrogen peroxide, and the absorbance was A_2_. In the damaged group, the volume was supplemented by distilled water without sugar solution, and the absorbance was A_1_. Vc was used as the positive control.
OH scavenging rate% = (A_3_ − A_1_)/(A_2_ − A_1_) × 100(1)

#### 2.3.2. Scavenging DPPH Free Radicals of GLPs

The experiment was slightly modified according to reference [16]. An amount of 1 mL of DPPH solution (0.4 mmol/L) dissolved in absolute ethanol was mixed with 3 mL of polysaccharide solution in the test tube, and the final concentrations of polysaccharide were 0.15–3.0 mg/mL, respectively. After mixing, they were kept away from light for 30 min at 25 °C, and then the absorbance was detected at 517 nm, with Vc as positive control.
DPPH scavenging rate% = [1 − (A_2_ − A_1_)/A_0_] × 100(2)

Among them, A_2_ is the absorbance of the mixture of DPPH solution and polysaccharide solution, A_1_ is the absorbance of the mixture of blank solvent (anhydrous ethanol) and polysaccharide solution, and A_0_ is the absorbance of the mixture of DPPH solution and anhydrous ethanol.

#### 2.3.3. Scavenging ABTS Free Radicals of GLPs

Referring to reference [17], ABTS solution (7 mmol/L) and K_2_S_2_O_8_ solution (2.45 mmol/L) were reacted at room temperature in the dark for 12–16 h at a volume ratio of 1:1. Then, 3 mL of mixed solution was reacted with 1 mL of polysaccharide solution (0.15–3 mg/mL) at 25 °C for 6 min, and the absorbance was measured at 734 nm.
ABTS scavenging rate% = [1 − (A_1_ − A_2_)/A_0_] × 100(3)

A_0_ is the control group without polysaccharide solution; A_1_ is the experimental group with polysaccharide solution; and A_2_ is the background group without ABST solution, that is, the absorbance A_2_ of the polysaccharide solution here is 0.

#### 2.3.4. Reducing Power of GLPs

The experiment was slightly modified according to reference [18]. The Prussian blue method was used to compare the reducing power of GLPs. An amount of 2.5 mL of polysaccharide solution at different concentrations (0.15–3.0 mg/mL) was added to the test tube, followed by PBS (2.5 mL) with pH = 6.6 and 1% K_4_[Fe(CN)_6_] (2.5 mL). It was then incubated at 50 °C for 30 min. After cooling to room temperature, 10% CCl_3_COOH (2.5 mL) was added and centrifuged at 3000 r/min for 10 min. An amount of 2.5 mL of supernatant was added to 5 mL of distilled water and 0.5 mL of 0.1% FeCl_3_·6H_2_O solution. After standing for 10 min, the absorbance was measured at 700 nm, and Vc was the positive control.

#### 2.3.5. Iron chelating ability of GLPs

According to reference [19], the experiment was slightly modified. First, 1.0 mL of polysaccharide solution with different concentrations (0.15–3 mg/mL) was added to the test tube, followed by ferrous chloride solution (2.0 mmol/L, 0.05 mL), phenazine solution (5.0 mmol/L, 0.2 mL) and water (2.25 mL). After 10 min of mixing, Fe^2+^ ions could form a stable chelate with phenazine, and their characteristic absorption at 562 nm could be detected using ultraviolet spectrophotometry. After adding polysaccharide solution, the content of Fe^2+^ in the solution decreased and the absorbance decreased due to the chelating of polysaccharide with Fe^2+^. The control group was treated with water instead of ferrous chloride solution, the blank group was treated with water instead of polysaccharide solution and EDTA was used as positive control.
Fe^2+^ chelating ability% = [A_0_ − (A_1_ − A_2_)]/A_0_ × 100(4)

A_1_ is the absorbance of polysaccharide solution group, A_2_ is the absorbance of the control group and A_0_ is the absorbance of the blank group.

### 2.4. GLPs Regulate Crystal Growth of Calcium Oxalate

#### 2.4.1. Crystal Growth

CaCl_2_ solution (22 mmol/L, 40 mL) was added to the beaker, and GLPs with a final concentration of 1.0 g/L were added. Then, 8 mL of distilled water was added and stirred at 37 °C for 5 min, and Na_2_Ox (22 mmol/L, 40 mL) was added to the above system. After reacting at 37 °C for 10 min, it was kept for 2 h and centrifuged. The bottom CaOx precipitate was washed with distilled water and anhydrous ethanol successively, then finally dried in a vacuum-drying oven.

#### 2.4.2. XRD Characterization of Crystals

According to the XRD spectrum, the relative contents of *COM* and *COD* in CaOx were calculated using the K value method. The percentage of *COD* is:$$COD\%=\frac{I_{COD}}{I_{COM}+I_{COD}}\times 100$$
where *I_COM_* is the intensity of the main diffraction peak ($\bar{1}01$) plane of *COM* and *I_COD_* is the intensity of the main diffraction peak (200) plane of *COD*.

#### 2.4.3. SEM Characterization of Crystals

First, 1 mg of CaOx crystal was ultrasonically dispersed in 10 mL of anhydrous ethanol. The crystal was spotted on the quartz substrate after low-power ultrasound for 3 min and dried in vacuum at 50 °C. The size and morphology of the crystal were observed using SEM after gold spraying.

#### 2.4.4. Zeta Potential of Crystals

An amount of 10 mg of CaOx crystals was dispersed in 30 mL of distilled water, and the Zeta potential value was measured using the Zeta potential analyzer after 10 min of ultrasound.

#### 2.4.5. Thermogravimetric Analysis of Crystals

A nitrogen gas flow was used, and the test temperature was 30–900 °C with a temperature rise rate of 10 °C/min.

### 2.5. Toxic Effects of Calcium Oxalate Crystals Regulated by GLPs on Cells

#### 2.5.1. Cell Culture

HK-2 cells were cultured with DMEM/F12 medium containing 10% FBS and 1% penicillin–streptomycin at 37 °C, 5% CO_2_ and saturated humidity. When the cells reached 80–90% confluence, 1 mL of 0.25% trypsin-EDTA digestive solution was added, and the cells were cultured in an incubator at 37 °C for 3–5 min. The degree of cell digestion was observed under a microscope. During moderate digestion, DMEM/F12 medium containing 10% FBS was added to terminate the digestion and gently blown to form a cell suspension. The cell suspension was inoculated into the corresponding culture dish to make the cells grow for subsequent experiments.

#### 2.5.2. CCK-8 Assay for the Toxicity of Different Crystals on HK-2 Cells

HK-2 cells were seeded in 96-well plates at 1.0 × 10^5^ cells/mL, 100 μL/well, and incubated in a 5% CO_2_ incubator at 37 °C for 24 h. After arrival time, the following groupings were performed: (1) normal control group: added serum-free medium; (2) pure COM and pure COD group: 200 μg/mL pure COM or pure COD crystals synthesized according to ref. [20] were added; (3) DC control group: added 200 µg/mL CaOx crystals formed without polysaccharides; (4) crystal damage group: added 200 µg/mL CaOx crystals regulated by GLPs. After incubation for 6 h, 10 μL of CCK-8 reagent was added to each well and incubated for 1 h at 37 °C in the dark. The *OD* value was detected using the microplate reader at 450 nm wavelength. The formula is as follows:$$Cell\;viability=\frac{treatment\;group(OD)}{control\;group(OD)}\times 100$$

#### 2.5.3. Cell SOD Activity

The cell inoculation density and grouping were the same as with CCK-8 detection. After arrival time, the cells were collected, and 20 μL of the sample, 160 μL of the NBT/enzyme working solution and 20 μL of the reaction initiation working solution were taken, with the OD value recorded as A_i_. The same volume of SOD detection buffer instead of the sample was taken as a blank control, and the OD value was recorded as A_1_. An amount of 40 μL of SOD detection buffer instead of 20 μL of the sample and 20 μL of the reaction initiation working solution was taken as a blank control, and the OD value was recorded as A_2_. After incubation at 37 °C for 30 min, the absorbance was measured at 560 nm, and 600 nm was set as the reference wavelength.

Calculation of inhibition percentage:inhibition percentage% = (A_1_ − A_i_)/(A_1_ − A_2_) × 100

Calculation of SOD activity:

SOD activity unit in the detection system = inhibition percentage/(1–inhibition percentage) units

#### 2.5.4. The Amount of MDA in Cells

The cell inoculation density and grouping were the same as with CCK-8 detection. After arrival time, cells were collected and 0.1 mL of different concentrations of standards were added to make standard curves. An amount of 0.1 mL of homogenate, lysate or PBS was added as a blank control, and 0.1 mL of samples were added for determination. Subsequently, 0.2 mL of MDA detection working solution was added to the above groups, mixed, heated at 100 °C for 15 min, cooled to room temperature, and centrifuged at 1000× *g* for 10 min. An amount of 200 μL of supernatant was added to the 96-well plate, and the absorbance was measured at 532 nm using an enzymometer.

#### 2.5.5. Detection of Osteopontin (OPN) Expression on Cell Surface

Cells were seeded in a confocal plate at 1.0 × 10^5^ cells/mL and 1 mL/well. The cell grouping was the same as with CCK-8 detection. After incubation for 6 h, the cells were washed with PBS, fixed with 4% paraformaldehyde for 10 min, washed with PBS three times and then blocked with goat serum for 20 min. Subsequently, the cells were incubated with OPN antibody (1:100) overnight at 4 °C, washed three times with PBS, incubated with FITC antibody (1:100) in a 37 °C incubator for 30 min in the dark, washed three times with PBS and finally incubated with a small amount of DAPI at room temperature for 10 min. The cells were washed three times with PBS, and then the expression of OPN was observed under a laser scanning confocal microscope. The fluorescence semi-quantitative analysis of OPN was performed using ImageJ software.

#### 2.5.6. Detection of Reactive Oxygen Species (ROS) Level by DCFH-DA Staining

Cells were seeded in a 6-well plate at 1.0 × 10^5^ cells/mL and 1 mL/well. Cell grouping was the same as with CCK-8 detection. After 6 h of incubation, the cells were washed with pre-cooled PBS, and 1 mL of DCFH-DA staining solution (1:1000) was added to each well. The cells were incubated at 37 °C for 30 min in the dark. The cells were washed with PBS and observed under an inverted fluorescence microscope. The fluorescence semi-quantitative analysis of ROS was performed using ImageJ software.

#### 2.5.7. Detection of Apoptosis and Necrosis by Hoechst 33342-PI Double Staining

The cell inoculation density and grouping were the same as with ROS detection. After incubation for 6 h, 800 μL of cell staining buffer was added to each well, and 5 μL of Hoechst 33,342 staining solution was added. Finally, 5 μL of PI staining solution was added and mixed well. After incubating at 4 °C for 30 min, the cells were washed with PBS and observed under an inverted fluorescence microscope. The fluorescence semi-quantitative analysis of apoptosis and necrosis was performed using ImageJ software.

#### 2.5.8. Statistical Analysis

All data are expressed as the mean ± standard deviation ($\bar{x}$ ± SD) of three parallel groups. A one-way ANOVA was performed using IBM SPSS Statistics 26 software. *p* > 0.05 indicates no significant difference, 0.01 < *p* < 0.05 indicates a significant difference and *p* < 0.01 indicates a highly significant difference.

## 3. Results

### 3.1. Degradation of GLP0

GLP0 was degraded by different concentrations of H_2_O_2_ to obtain seven degraded polysaccharide fragments (Table 1). They were named GLP1–GLP7, with molecular weights of 106, 49.6, 10.5, 6.14, 5.06, 3.71 and 2.42 kDa, respectively.

### 3.2. Antioxidant Activity of GLPs

#### 3.2.1. Scavenging Hydroxyl Radical (·OH)

The hydroxyl radical is a free radical that does great harm to organisms, polysaccharides can eliminate them by providing single electrons or hydrogen atoms for free radicals [21]. Figure 1A shows the hydroxyl radical scavenging ability of the seven polysaccharides (named GLP1, GLP2, GLP3, GLP4, GLP5, GLP6 and GLP7, respectively) with *M*_w_ of 2.42, 3.71, 5.06, 6.14, 10.5, 49.6 and 106 kDa, respectively. It can be seen that:

(1) For the same kind of polysaccharide, with the increase in polysaccharide concentration, the ability of scavenging hydroxyl radicals increased, indicating that the antioxidant activity of polysaccharide was concentration-dependent.

(2) For different kinds of polysaccharides, when the *M*_w_ of polysaccharides increased from 2.42 kDa to 49.6 kDa, the scavenging ability of polysaccharides on hydroxyl radicals increased continuously. However, when the *M*_w_ reached 106 kDa, its scavenging ability decreased. GLP2, with *M*_w_ = 49.6 kDa, had the strongest hydroxyl radical scavenging ability.

#### 3.2.2. Scavenging DPPH Free Radical

The DPPH free radical is a stable free radical that is commonly used to test antioxidant capacity. The DPPH radical has maximum absorbance at 517 nm. When antioxidants provide hydrogen atoms for the DPPH radical to obtain the non-radical compound DPPH-H, the purple disappears and the absorbance decreases [22]. The results of scavenging DPPH free radicals by seven GLPs are shown in Figure 1B. For the same kind of polysaccharide, its DPPH free radical scavenging ability was concentration-dependent.

The scavenging rule of the DPPH free radical by different kinds of polysaccharides was consistent with that of the hydroxyl free radical. That is, GLP2 with *M*_w_ = 49.6 kDa had the strongest scavenging ability.

#### 3.2.3. Scavenging ABTS Free Radical

ABTS free radical scavenging is a common method for detecting the total antioxidant capacity of antioxidants [23]. As shown in Figure 1C, the ability of the same kind of polysaccharide to scavenge ABTS free radicals was concentration-dependent. The higher the polysaccharide concentration was, the stronger the scavenging capacity. For different kinds of polysaccharides, the strongest scavenging ability was still GLP2, with *M*_w_ = 49.6 kDa.

#### 3.2.4. Reducing Power

The antioxidant can provide a single electron to reduce the trivalent iron of K_4_[Fe(CN)_6_]) to divalent iron, which further reacts with FeCl_3_·6H_2_O to form Prussian blue (Fe_4_[Fe_6_(CN)_3_]_3_), which has maximum absorbance at 700 nm. Therefore, the reducing power can be indirectly reflected by measuring the absorbance at 700 nm. The greater the absorbance, the stronger the reducing power of polysaccharides.

From Figure 1D, it can be seen that the absorbance of the same kind of polysaccharide at 700 nm increased with the increase in polysaccharide concentration; that is, the reducing power increased with the increase in polysaccharide concentration and was shown to be concentration-dependent.

For different kinds of polysaccharides, with the increase in *M*_w_ from 2.42 kDa to 49.6 kDa, the absorbance of polysaccharides increased, indicating that the reducing power was enhanced. However, the absorbance of GLP1 with *M*_w_ = 106 kDa was lower than that of GLP2, with *M*_w_ = 49.6 kDa, indicating that GLP had the best reducing power when *M*_w_ was about 49.6 kDa.

#### 3.2.5. Fe^2+^ Chelating Ability

Ferrous ions (Fe^2+^) can activate lipid peroxidation and accelerate the oxidation rate of lipid compounds through the Fenton reaction, so their antioxidant activity can be evaluated by detecting the ability of polysaccharides to chelate Fe^2+^ ions [24].

Figure 1E shows that GLPs have the ability to scavenge Fe^2+^ in a concentration-dependent manner. At the same concentration, the chelating ability of GLPs to Fe^2+^ increased with the decrease in *M*_w_. For example, at the concentration of 2.0 mg/mL, the Fe^2+^chelation rate of was GLP7 (86.3%) > GLP6 (85.9%) > GLP5 (84.5%) > GLP4 (79.8%) > GLP3 (55.5%) > GLP2 (46.6%) > GLP1 (41.4%).

### 3.3. GLPs Regulate Crystal Growth of Calcium Oxalate

According to the preliminary experimental results, four representative polysaccharides, GLP1, GLP3, GLP5 and GLP7, with relatively large differences in properties and *M*_w_ of 106, 10.5, 5.06 and 2.42 kDa, were selected to study their regulatory effects on the growth of CaOx crystals.

#### 3.3.1. XRD Characterization

Figure 2 shows the XRD spectra of CaOx crystals regulated by GLPs at 1.0 g/L. The diffraction peaks of crystals without polysaccharides appeared at d = 0.591, 0.364, 0.296 and 0.235 nm, which are the characteristic diffraction peaks of COM crystals, corresponding to their ($\bar{1}01$), (020), ($\bar{2}02$) and (130) crystal planes; that is only COM crystals were formed in the crystal control group without polysaccharides (Figure 2A).

In the CaOx crystals regulated by GLPs, with the decrease in *M*_w_, the diffraction peaks at d = 0.617, 0.441 and 0.277 nm in the XRD spectra gradually increased. These diffraction peaks corresponded to the (200), (211) and (411) crystal planes of COD crystals (Figure 2A), indicating that the proportion of COD crystals in CaOx crystals increased with the decrease in *M*_w_. The semi-quantitative calculation of the K value method showed that the percentages of COD regulated by GLP1, GLP3, GLP5 and GLP7 were 16.67%, 60.87%, 67.84% and 79.52%, respectively (Figure 2B).

#### 3.3.2. SEM Observation

Figure 3 shows the SEM images of CaOx crystals regulated by four GLPs at a concentration of 1.0 g/L. In the crystal group without polysaccharides, the crystal morphology was flaky and the crystal size was small (about 0.8 μm). The crystal morphology was irregular, and there were obvious aggregation phenomena (Figure 3a). XRD (Figure 2A) shows that these crystals were COM crystals. In the presence of 1.0 g/L GLPs, straw-hat-like or tetragonal bipyramidal crystals appeared. These crystals were COD crystals, which were consistent with the morphology of COD reported in the literature [25]. The crystal size regulated by polysaccharides was larger than that of the control group (about 1.6–3.7 μm). As the *M*_w_ of polysaccharides decreased (from GLP1 to GLP7), the proportion of straw-cap COD crystals increased, and the degree of crystal aggregation decreased significantly.

#### 3.3.3. Zeta Potential

Compared with the crystals without polysaccharides (–1.56 mV), the Zeta potential of the crystals regulated by the four GLPs became more negative (–21.7 mV to –30.2 mV) (Figure 4). With the decrease in *M*_w_ of GLPs, the Zeta potential became more negative, indicating that the lower the *M*_w_ of GLPs, the stronger the anti-aggregation ability of CaOx crystals regulated by GLPs.

#### 3.3.4. Thermal Gravimetric Analysis

The decomposition of COM, namely CaC_2_O_4_·H_2_O, went through three stages, and the theoretical weight loss rates were 12.33% (Formula (5)), 19.18% (Formula (6)) and 30.13% (Formula (7)), respectively. It can be seen from Figure 5 that the crystal decomposition without polysaccharides also went through three stages (DC group in Figure 5), and the weight loss rates were 12.82%, 18.42% and 29.65%, respectively (Table 2), which was basically consistent with the theoretical weight loss rate of COM, indicating that the crystal control group without polysaccharides was COM.
CaC_2_O_4_·H_2_O → CaC_2_O_4_ + H_2_O(5)
CaC_2_O_4_ → CO + CaCO_3_(6)
CaCO_3_ → CO_2_ + CaO(7)

In the CaOx crystals regulated by four GLPs at a concentration of 1.0 g/L, COM and COD crystals were formed simultaneously, and GLPs may also be adsorbed in the regulated crystals, which results in different TGA curves to the DC group.

With the decrease in GLPs *M*_w_, the weight loss rate in stage I (30–220 °C) increased gradually (13.68–17.50%) (Table 2). This is because more and more COD crystals were regulated by GLPs, and COD crystal had one more bound water than COM crystal. Therefore, the weight loss rate at stage I was getting higher and higher, which is consistent with the above XRD (Figure 2) and SEM (Figure 3) results.

The crystals obtained without polysaccharides did not show weight loss at stage II (220–400 °C), whereas the crystals regulated by GLPs showed weightlessness at this temperature, which was caused by the decomposition of polysaccharides [26]. In other words, the mass loss within the temperature range can be considered as the mass of polysaccharides adsorbed in the crystals. As the *M*_w_ of GLPs decreased, the proportion of polysaccharides adsorbed into the crystal increased (5.04–9.49%) (Table 2). The adsorption of polysaccharides on the crystal surface is mainly attributed to the special interaction between polysaccharides and crystals [27]. Fang et al. [28] showed that polysaccharides rich in –OSO_3_^–^ had a high interaction with CaOx crystals, and thus, more polysaccharides could be embedded into CaOx crystals, which is consistent with the results of this paper.

In stage III (440–540 °C), the weight loss rate decreased from 18.42% in the DC group to 9.93–13.59% in the GLPs group (Table 2). With the decrease in GLPs *M*_w_, the weight loss rate was smaller, which was due to the higher proportion of crystalline water and polysaccharide loss in stage I and II.

In stage IV of decomposition (540–740 °C), with the decrease in GLPs *M*_w_, the weight loss rate also decreased to varying degrees (26.40–29.47%) (Table 2), which was consistent with the above results.

### 3.4. Cytotoxicity of CaOx Crystals Regulated by GLPs with Different Molecular Weights

The cytotoxicity of CaOx crystals regulated by GLP1, GLP3, GLP5 and GLP7 was studied. Due to the different properties of polysaccharides, the properties of the regulated CaOx crystals are also different, thus showing different cytotoxicity.

#### 3.4.1. Cell Viability

Figure 6 shows the cell viability changes of normal HK-2 cells after injury induced by pure COM crystal, pure COD crystal, a crystal control group without polysaccharides (DC group) and CaOx crystals regulated by GLP1, GLP3, GLP5 and GLP7. Compared with the normal control group (100% ± 2.4%), the cell viability of the crystal damage group was significantly decreased (56.5–86.8%), indicating that CaOx crystals caused damage to cells. The cytotoxicity of the crystals regulated by GLPs (65.3–86.8%) was weaker than that of the DC group (56.5%), and the cytotoxicity of CaOx crystals decreased with the decrease in *M*_w_ of GLPs.

The cytotoxicity of pure COM was the highest (with cell viability of 56.5%), and the cytotoxicity of pure COD (with cell viability of 78.8%) was greater than that of the GLP7 group (with cell viability of 86.8%). Even the GLP7 group only regulated 79.52% of COD formation due to the embedding of polysaccharides in the latter (Figure 5) reducing cytotoxicity [29].

#### 3.4.2. SOD Activity and MDA

Compared with the normal control group (9.4 U/mL) with higher SOD activity, the SOD activity of HK-2 cells in the crystal group without polysaccharides (DC group) (3.48 U/mL) was significantly decreased, which was attributed to the large number of cell deaths caused by crystal damage. The CaOx regulated by different GLPs could also cause some cell death, but the death rate was lower than that of the DC group. Therefore, the SOD activity of the GLPs groups (4.11–6.68 U/mL) was higher than that of the DC group (Figure 7a), and the SOD activity increased with the decrease in *M*_w_ of GLPs.

The content of MDA in the crystal damaged group (3.50–8.03 nmol/mL) was higher than that in the normal control group (1.22 nmol/mL) (Figure 7b). MDA in the GLPs group decreased with the decrease in GLPs *M*_w_. Figure 7 further shows that CaOx crystals have toxic effects on normal HK-2 cells.

#### 3.4.3. Osteopontin (OPN) Expression

Immunofluorescence was used to detect the expression of OPN in HK-2 cells after CaOx crystal injury (Figure 8). In the normal control group, there was only a small amount of OPN expression on the cell surface, which was reflected as weak green fluorescence on the cell surface (Figure 8A). The green fluorescence on the cell surface of the crystal damage group was significantly enhanced, and the fluorescence semi-quantitative analysis showed that the average fluorescence intensity was significantly higher than that of the normal control group (371.9–589.9%) (Figure 8B), indicating that the expression of OPN was significantly up-regulated. The fluorescence of the GLP7 group, with the smallest *M*_w_, was relatively weak (371.9%).

#### 3.4.4. ROS Level

Figure 9 shows the effect of CaOx crystals on the ROS level in HK-2 cells detected by DCFH-DA. Compared with the low ROS level in the normal control group, the green fluorescence of the cells in the crystal damage group was enhanced to varying degrees (Figure 9A), indicating that the ROS level in the damage group was increased. The ROS level in the crystal control group without polysaccharides (861.4%) was higher than that in the GLPs group (304.2–775.4%) (Figure 9B), while the ROS level in the GLP1 group with the largest *M*_w_ (775.4%)was significantly higher than that in the GLP7 group with the smallest *M*_w_ (304.2%).

#### 3.4.5. Apoptosis and Necrosis Detected by Hoechst 33342-PI Double Staining

The effects of CaOx crystals on apoptosis and necrosis of HK-2 cells were detected using Hoechst 33,342 staining-PI double staining. The number of red-stained cells in the crystal damage group was significantly higher than that in the normal control group (Figure 10A), indicating that the cells in the crystal damage group showed different degrees of late apoptosis or necrosis. The number of red-stained cells in the GLPs crystal group (325.6–473.8%) was positively correlated with the polysaccharide *M*_w_ (Figure 10B), i.e., the smaller the *M*_w_, the fewer the red-stained cells and the less the cell damage.

## 4. Discussion

### 4.1. Effect of Polysaccharide Molecular Weight on Its Antioxidant Activity

The factors affecting the biological activity of polysaccharides are diverse, such as monosaccharide composition, main chain composition, branching degree and branched chain, conformation, molecular weight and acid group content in polysaccharides. From the results of ^1^H NMR, ^13^C NMR and GC-MS [14], it can be seen that the degradation of GLP0 by H_2_O_2_ does not change the main skeleton structure of polysaccharides (all the polysaccharides composed of β-D-galactose and 6-O-sulfate-3,6-α-L-galactose) and monosaccharide composition. The groups that attach to the carbohydrate part, that is, the content of acidic groups (–OSO_3_^–^ and –COO^–^) in the GLPs (13.07–13.56%) is also similar (Table 1). Therefore, the molecular weight *M*_w_ of GLPs is the main factor affecting the antioxidant activity of GLPs and regulating the growth of CaOx crystals [12,30]. For polysaccharides with different properties, the molecular weight ranges in which they exhibit optimal bioactivity may be different. Too large or too small *M*_w_ will lead to reduced antioxidant activity of polysaccharides. In this experiment, the *M*_w_ of GLPs had the best antioxidant effect when it was about 49.6 kDa (GLP2). When the *M*_w_ was less than or greater than 49.6 kDa, its antioxidant effect gradually decreased. The reasons are as follows:

#### 4.1.1. The Antioxidant Activity of Polysaccharides Decreased When M_w_ Was Too Large

The results in Figure 1 show that GLP2 with *M*_w_ = 49.6 kDa had the strongest free radical scavenging (·OH, DPPH and ABTS) (Figure 1A–C) and reducing power (Figure 1D). The reducing power of polysaccharides had a direct positive correlation with its antioxidant capacity. The stronger the reducing power was, the stronger the antioxidant capacity was [30,31]. The antioxidant activity of GLPs with *M*_w_ greater than or less than 49.6 kDa was weakened. When the polysaccharide *M*_w_ was too large, not only was its water solubility low, the viscosity of the solution was large, which inhibited its activity. In addition, the polysaccharide with a too-large *M*_w_ had a closer structure and stronger intramolecular hydrogen bonds, resulting in lower activity of active groups and weaker antioxidant capacity [32]. Qi et al. [33] studied the antioxidant activity of four different *M*_w_ sulfated *Ulva pertusa* Kjellm (Chlorophyta) (U, U1, U2 and U3, *M*_w_ were 151.7, 64.5, 58.0 and 28.2 kDa, respectively). Among them, the smallest *M*_w_ U3 had the strongest scavenging superoxide anion and hydroxyl radical activity, and its IC_50_ were 22.1 μg/mL and 2.8 mg/mL, respectively. The reduction ability and chelating ability of U3 were also the strongest in the four samples.

#### 4.1.2. The Antioxidant Activity of Polysaccharides Decreased When Molecular Weight Was Too Small

However, it is not that the smaller the *M*_w_ of polysaccharides is, the greater its activity is. When *M*_w_ < 49.6 kDa, the antioxidant activity of GLPs decreased with the decrease in *M*_w_. Polysaccharides with repetitive structure and more electron donors (such as hydroxyl) can provide a large number of single electrons and protons for free radicals to terminate free radical chain reactions. When the *M*_w_ of polysaccharides was too small, the unique bond linkage mode of polysaccharides and the stereo structure (i.e., conformation) of polysaccharides formed based on intramolecular hydrogen bonds were destroyed, resulting in a decrease in the biological activity of polysaccharides. Lv et al. [34] isolated two polysaccharides (PMP-1 and PMP-2) with *M*_w_ of 480 and 610 kDa from *Polygonum multiflorum* Thunb using column chromatography. The IC_50_ values of PMP-1 for scavenging superoxide anion, hydroxyl and hydrogen peroxide were 1.41, 2.65 and 1.39 mg/mL, respectively, while those of PMP-2 were only 0.47, 0.93 and 0.60 mg/mL, respectively, indicating that the antioxidant activity of the PMP-1 component with small *M*_w_ decreased. Chen et al. [35] extracted three polysaccharide components, LA, LB and LC, from Lentinus edodes, with *M*_w_ of 150, 220 and 290 kDa, respectively. In the concentration range of 0.25–4 mg/mL, their scavenging ability for the hydroxyl radical and chelating ability to Fe^2+^ were LA < LB < LC, indicating that the antioxidant capacity of the LA component with the smallest *M*_w_ was the weakest. The extracellular polysaccharide of *Saccharomyces cerevisiae* (sGSCs) had lower biological activity at *M*_w_ = 5–10 kDa, but higher activity at 100–200 kDa [36].

#### 4.1.3. The Antioxidant Activity of Polysaccharides with Moderate Molecular Weight Was the Highest

In this study, GLPs *M*_w_ at about 49.6 kDa (GLP2) had the best antioxidant effect; when *M*_w_ was less than or greater than 49.6 kDa, the antioxidant effect was gradually reduced. When the *M*_w_ of polysaccharides is moderate, it can not only retain enough sugar units to form the three-dimensional structure of polysaccharides to ensure that its conformation is not destroyed, but also break the highly close molecular structure of natural large *M*_w_ polysaccharides before degradation so that the active functional groups (such as sulfuric acid group, carboxyl group and hydroxyl group) are exposed, showing the maximum degree of freedom. The spatial steric hindrance is also the smallest when reacting with organisms, and the water solubility is increased, thus exerting the maximum biological activity [37]. In general, the ability of antioxidants to scavenge ABTS free radicals is positively correlated with the ability to scavenge DPPH free radicals. However, GLPs with different *M*_w_ had stronger ABTS radical scavenging ability than DPPH radical scavenging ability. For example, at the concentration of 3.0 mg/mL, the ABTS radical scavenging rate of GLP2 was 91.0%, while the DPPH radical scavenging rate was only 47.9%. This is because the ABTS reagent is more suitable for hydrophilic antioxidants, and the DPPH reagent is more suitable for hydrophobic antioxidants [38]. GLPs are hydrophilic polysaccharides, and GLPs have a higher ABTS radical scavenging rate. Sheng et al. [39] degraded CPA with hydrogen peroxide (*M*_w_ = 16.89 kDa) and obtained four kinds of degradation sugars (CPA-1, CPA-2, CPA-3 and CPA-4) with *M*_w_ of 14.53, 12.37, 11.55 and 0.64 kDa, respectively. At the concentration of 200 μg/mL, the scavenging rates of CPA, CPA-1, CPA-2, CPA-3 and CPA-4 on superoxide anion were 9.68%, 7.81%, 10.71%, 8.97% and 8.57%, respectively. It indicated that only CPA-2 with moderate *M*_w_ had the strongest free radical scavenging ability. The results of this paper are consistent with the results of Sheng et al. [39].

#### 4.1.4. The Ability of Small M_w_ Polysaccharide to Chelate Fe^2+^Ions Was Stronger

The results for Fe^2+^ chelating ability of GLPs with different *M*_w_ were not completely consistent with the free radical scavenging and reducing power. With the decrease in *M*_w_ of GLPs, their ability to chelate Fe^2+^ ions was continuously enhanced (Figure 1E). This is significantly different from the pattern of GLPs’ scavenging free radicals (·OH, DPPH and ABTS) and reducing power (Figure 1F). This is because the smaller the *M*_w_ is, the more exposed the active groups (–OSO_3_^–^ and –COOH, etc.) it contains and the greater the degree of freedom it has, and thus, the smaller the steric hindrance when it chelates with Fe^2+^ ions. Much of the literature also shows that low *M*_w_ polysaccharide has a stronger ability with complex Fe^2+^ ions. For example, Zou et al. [40] tested the chelating ability of three polysaccharides with *M*_w_ of 13,500, 11,700 and 11,500 Da with Fe^2+^ ions. When the polysaccharide concentration was 500 μg/mL, the chelating rates with Fe^2+^ ions were 2.47%, 5.48% and 5.96%, respectively. That is, the smaller the *M*_w_, the stronger the chelating ability of polysaccharides with Fe^2+^ ions.

### 4.2. Regulation Ability of Different GLPs on CaOx Crystal Growth

#### 4.2.1. GLPs Induce the Formation of COD Crystals

XRD (Figure 2A) and SEM (Figure 3) showed that the crystal control group without polysaccharides only formed COM crystals, while each GLP could regulate the formation of COD. As GLPs *M*_w_ decreased, the percentage of COD increased (Figure 2B). This is because GLPs contain –OSO_3_^–^ with negative charge, while the ($\bar{1}01$) crystal plane of COM crystal has positive charge, and the adsorption sites on the surface of COM crystal are more than those on the surface of COD crystal [41]. Therefore, GLPs can be adsorbed on the surface of COM crystals by electrostatic interaction, thereby preventing the deposition of Ca^2+^ ions on the surface of COM crystals and inhibiting the growth of COM crystals. In contrast, the COD surface is electrically neutral. Therefore, GLPs inhibit COM growth while promoting the formation of COD [42]. Since the adhesion of COD crystals to renal epithelial cells is much weaker than that of COM, COD is more easily excreted than COM; that is, inducing the formation of COD is more likely to reduce the risk of CaOx kidney stone formation than inducing the formation of COM.

#### 4.2.2. GLPs Inhibit Crystal Aggregation

SEM also showed that GLPs inhibited crystal aggregation (Figure 3). Compared with the crystal control group without polysaccharides, the dispersion of CaOx crystals regulated by GLPs was higher. Because GLPs can be adsorbed on the crystal surface in the presence of GLPs, the absolute value of Zeta potential on the crystal surface increases (Figure 4). It is already known that the size of the Zeta potential can reflect the mutual repulsion or attraction between particles. The higher the absolute value of Zeta potential, the more stable the system; the lower the absolute value of Zeta potential, the more likely the system is to condense or agglomerate. The results of this study show that as the *M*_w_ of GLPs decreased, the Zeta potential of the crystal surface became more negative; the repulsion between crystals was larger, and the crystals were more dispersed.

### 4.3. Toxicity Difference of CaOx Crystals Regulated by GLPs on HK-2 Cells

The results of this study show that the cytotoxicity of CaOx crystals regulated by GLPs with different *M*_w_ was negatively correlated with the antioxidant activity of GLPs; that is, the smaller the *M*_w_, the stronger the antioxidant activity of GLPs and the smaller the cytotoxicity of CaOx crystals regulated by GLPs. This is mainly due to the difference in the percentage of COM and COD in CaOx crystals regulated by different GLPs. Under the same concentration condition, the GLP1 with the largest *M*_w_ only regulates 16.67% of COD, while the GLP7 with the smallest *M*_w_ regulates 79.52% of COD (Figure 2B). Since the positive charge density on the COM surface is higher than that on COD [43], and the specific surface area of COM crystal is also larger than that of COD (Figure 3), the adhesion of COM to damaged cells with negatively charged molecules is stronger than that of COD [44]; that is, the cytotoxicity of COM is much larger than that of COD [20,43]. The results of this paper show that the higher the COD content of the crystal, the greater the cell viability (Figure 6) and MDA level (Figure 7b), and the smaller the SOD activity (Figure 7a), OPN expression (Figure 8), ROS level (Figure 9) and necrotic cell rate (Figure 10); that is, the cytotoxicity of CaOx crystals regulated by low *M*_w_ GLPs is less than that of CaOx crystals regulated by high *M*_w_ GLPs. Figure 11 shows the different cytotoxicity of CaOx crystals regulated by two different *M*_w_ polysaccharides (GLP1 and GLP7) on HK-2 cells. This result relates to the following factors:

#### 4.3.1. The Cytotoxicity of COM Is Greater Than That of COD

With the decrease in *M*_w_ of GLPs, the percentage of COD in the regulated crystals increased (Figure 2B). Since the cytotoxicity of COD crystal is lower than that of COM crystal [45], the higher the percentage of COD in CaOx crystal, the lower the cytotoxicity. In addition, COM and COD crystals have different dynamic binding modes and pathological behavior [46]. COM has a strong binding ability for binding to the cell surface of different organic molecular arrays, thus becoming an aggregation center for stone formation [47]; COD has difficulties forming stable aggregates or strong adhesion contact with renal epithelial cells [48]. Therefore, compared with COM crystals, COD crystals cause less damage to renal tubular epithelial cells [49]. In fact, there are a large number of COD microcrystals in urine, and the proportion of COM in stones is high [48].

#### 4.3.2. Crystals with Blunt Morphology Have Less Cytotoxicity

It can be seen from Figure 3 that as the *M*_w_ of GLPs decreases, the regulated CaOx crystal edges become more blunt. This is because there is a complexation–dissociation equilibrium between GLPs and Ca^2+^ ions on the surface of CaOx crystals (especially Ca^2+^ ions located at the edge and tip of CaOx crystals) during the growth of CaOx crystals regulated by GLPs. On the one hand, the formed crystals dissolve continuously due to complexation with GLPs. On the other hand, the dissolved Ca^2+^ ions are continuously deposited on the surface of CaOx crystals. This continuous dissolution–deposition equilibrium leads to more round and blunt crystals [49]. GLP7 has the smallest *M*_w_ and the highest activity, so it has the strongest complexation–dissociation equilibrium balancewith Ca^2+^ ions. The regulated crystals not only have the highest COD content, but also are more round and blunt, which can reduce the mechanical damage to cells. In addition, blunt crystals are more easily excreted through urine, which helps reduce the risk of kidney stones [29].

#### 4.3.3. Small-Size Crystals Have Greater Cytotoxicity

Furthermore, the toxicity of COM and COD to cells is also closely related to the size of crystals. It can be seen from Figure 3 that the size of CaOx crystals increased with the decrease in GLPs *M*_w_, but the crystals regulated by the four GLPs groups were larger than the control group without polysaccharides. When the crystal size increased, the cytotoxicity decreased [50].

#### 4.3.4. Crystal Toxicity with High Dispersion Is Smaller

In addition, the degree of crystal aggregation also has an important effect on cytotoxicity. Aggregated crystals are more likely to adhere to the surface of renal epithelial cells and damage cells, which also increases the risk of kidney stones formation [51]. In fact, compared with healthy controls, urinary microcrystals in patients with kidney stones showed sharp edges and obvious aggregation [20].

Since the formed CaOx crystals are highly aggregated without GLPs (Figure 3), their cytotoxicity is also the largest (Figure 6). With the decrease in *M*_w_ of GLPs, the aggregation degree of the regulated CaOx crystals decreased, and the cytotoxicity of crystals also decreased.

#### 4.3.5. The Crystal Cytotoxicity of Doped Polysaccharides Is Smaller

Crystal surface adsorption or crystal-doped polysaccharides can reduce the toxicity of crystals to cells. Our previous studies [52,53] have shown that a certain amount of polysaccharide (6–21%) is doped or adsorbed in the CaOx crystals generated by polysaccharide regulation. In this paper, a similar conclusion was obtained through thermogravimetric analysis. The polysaccharide contents in the crystals regulated by GLP1, GLP3, GLP5 and GLP7 were 5.04%, 6.31%, 8.81% and 9.49%, respectively (Table 2). The polysaccharides doped in the crystals can play a good role in cell protection and reduce the toxicity of CaOx crystals, including preventing the destruction of the cell morphology and cytoskeleton, inhibiting the production of ROS and the decrease in lysosome integrity and reducing the expression of OPN and transmembrane proteins (CD44).

## 5. Conclusions

The seven kinds of GLPs, with *M*_w_ of 106, 49.6, 10.5, 6.14, 5.06, 3.71 and 2.42 kDa, had good reducing power and the ability to scavenge hydroxyl radical, DPPH radical and ABTS radical. GLP2, with *M*_w_ of 49.6 kDa, had the best antioxidant effect. With the decrease in *M*_w_ of GLPs, its ability to chelate Fe^2+^ ions increased. These GLPs can inhibit the growth of COM crystals to varying degrees, induce the formation of COD crystals, improve the absolute value of Zeta potential on the crystal surface, reduce the aggregation between crystals and make the edges and corners of the crystals more blunt. The toxicity of CaOx crystals regulated by GLPs on HK-2 cells was reduced, resulting in increased cell viability and MDA level, as well as decreased SOD activity, OPN expression, ROS level and necrotic cell number. GLPs, especially GLP7, may be potential stone preventive drugs.

## Figures and Tables

**Figure 1 foods-12-01031-f001:**
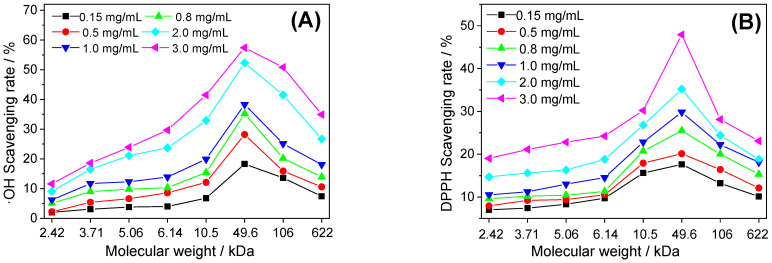
Antioxidant activities of *Gracilaria lemaneiformis* polysaccharides with different molecular weights. (**A**) Scavenging ·OH free radicals; (**B**) Scavenging DPPH radical; (**C**) Scavenging ABTS free radicals; (**D**) Reducing power; (**E**) Fe^2+^ chelating ability; (**F**) Comparison of antioxidant activity of each polysaccharide at 3.0 mg/mL. Values are means ± S.D (*n* = 3).

**Figure 2 foods-12-01031-f002:**
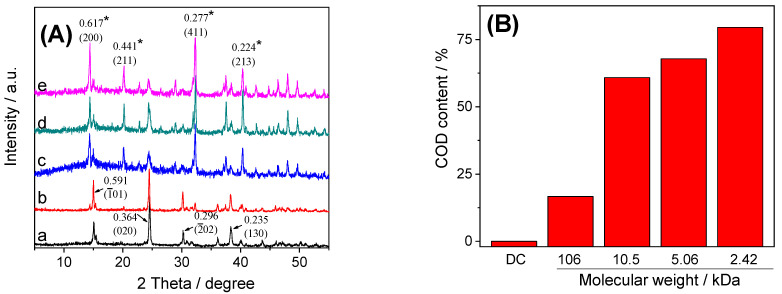
XRD spectrum (**A**) and percentage content of COD in CaOx crystals (**B**) regulated by GLPswith concentration of 1.0 g/L. (a) Crystal control group without polysaccharides; (b) GLP1; (c) GLP3; (d) GLP5; (e) GLP7. The crystal plane with asterisk shows COD, and the crystal plane without asterisk shows COM. The *M*_w_ of GLP1, GLP3, GLP5 and GLP7 was 106, 10.5, 5.06 and 2.42 kDa respectively.

**Figure 3 foods-12-01031-f003:**
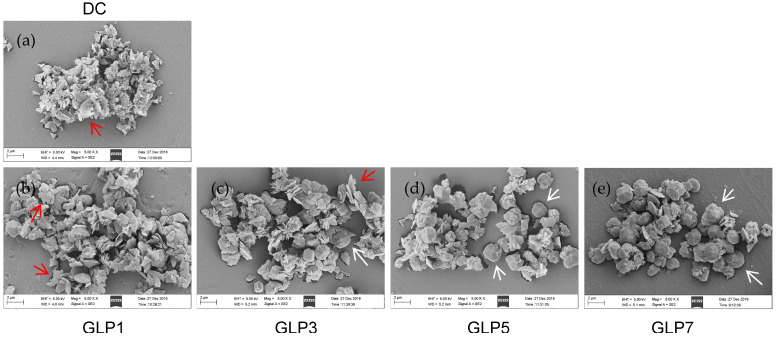
SEM images of the CaOx crystals regulated by GLPs with concentration of 1.0 g/L. (**a**) Crystal control group without polysaccharides; (**b**) GLP1; (**c**) GLP3; (**d**) GLP5; (**e**) GLP7. *c* (CaOx) = 10 mmol/L. The red arrows refer to COM crystals, and the white arrows refer to COD crystals.

**Figure 4 foods-12-01031-f004:**
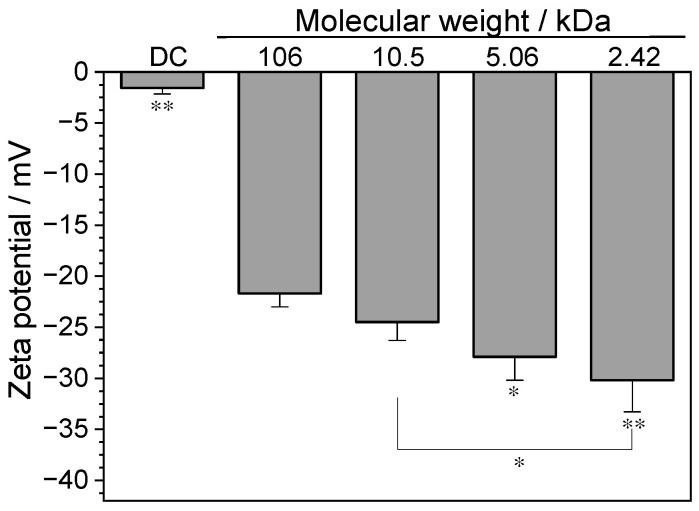
Zeta potential of the CaOx crystals regulated by GLPs with concentration of 1.0 g/L. Compared with DC, * *p* < 0.05; ** *p* < 0.01.

**Figure 5 foods-12-01031-f005:**
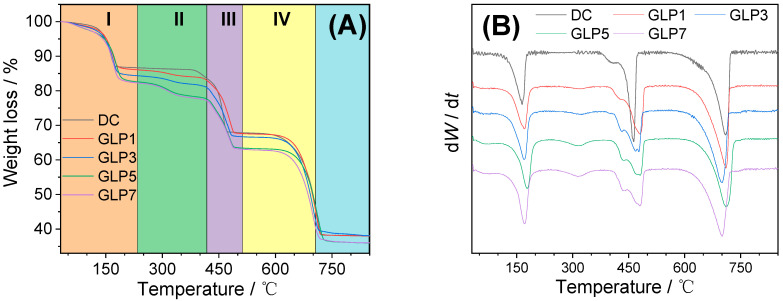
TGA (**A**) and DTG (**B**) curves of the CaOx crystals regulated by GLPs with concentration of 1.0 g/L. Stage I corresponds to the loss of bound water in COM or COD; Stage II corresponds to the loss of polysaccharides; Stage III corresponds to the loss of CO in CaC_2_O_4_ and Stage IV corresponds to the loss of CO_2_ in CaCO_3_.

**Figure 6 foods-12-01031-f006:**
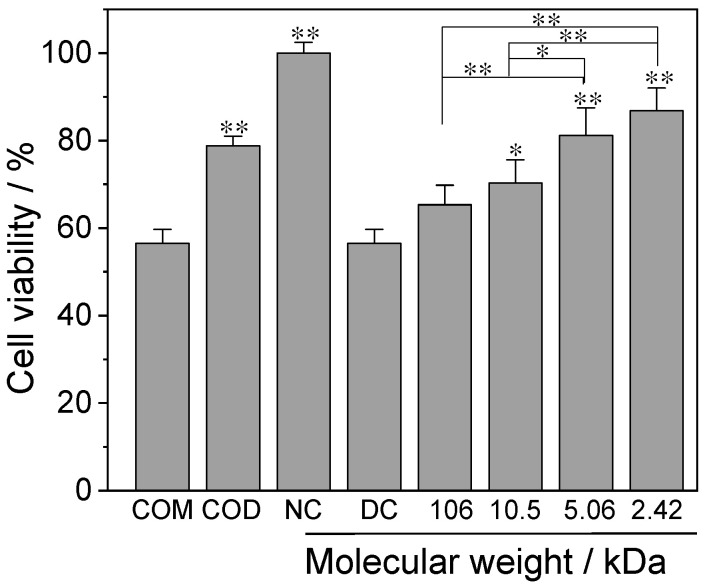
Effects of CaOx crystals regulated by GLPs at a concentration of 1.0 g/L on the viability of HK-2 cells. Crystal concentration: 200 μg/mL; time: 6 h. NC: normal control group; DC: crystal control group without polysaccharides. Compared with DC, * *p* < 0.05; ** *p* < 0.01.

**Figure 7 foods-12-01031-f007:**
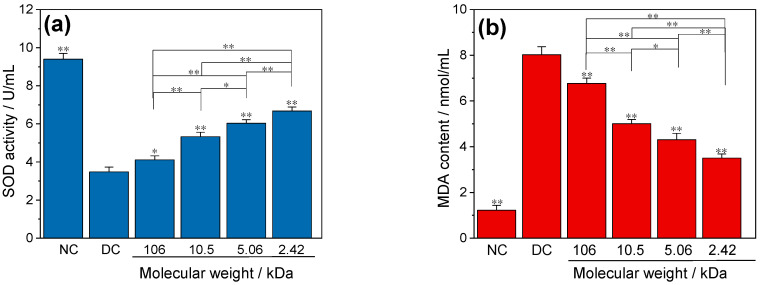
Effects of CaOx crystals regulated by GLPs at a concentration of 1.0 g/L on the SOD activity (**a**) and the amount of MDA (**b**) in HK-2 cells. GLPs concentration: 1.0 g/L. Compared with DC, * *p* < 0.05; ** *p* < 0.01.

**Figure 8 foods-12-01031-f008:**
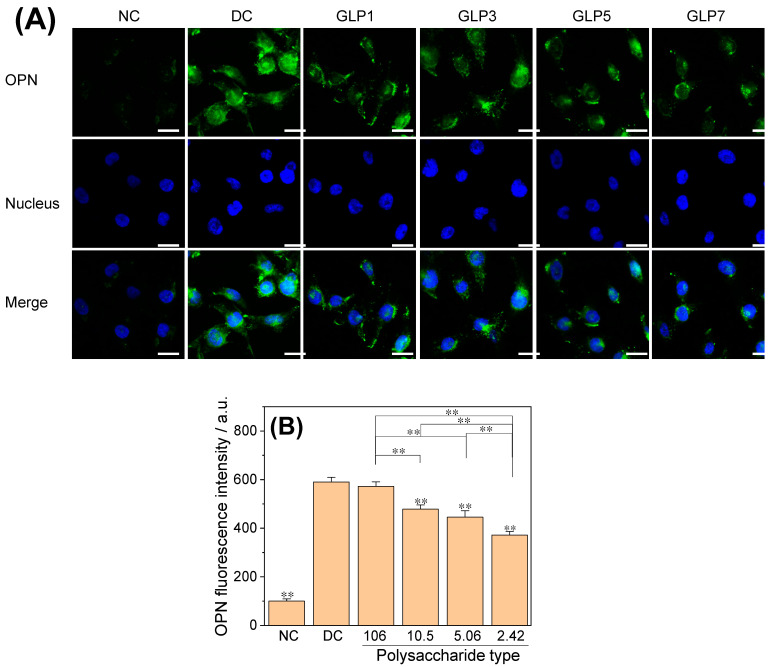
Effects of CaOx crystals regulated by GLPs at a concentration of 1.0 g/L on the expression of OPN molecules on the surface of HK-2 cells. Scale: 20 μm. In the figure, blue fluorescence represents the nucleus, and green fluorescence represents the OPN molecule. (**A**) OPN fluorescence map; (**B**) fluorescence intensity quantitative diagram. The *M*_w_ of GLP1, GLP3, GLP5 and GLP7 was 106, 10.5, 5.06 and 2.42 kDa, respectively. NC: normal control group; DC: crystal control group without polysaccharides. Compared with DC, ** *p* < 0.01.

**Figure 9 foods-12-01031-f009:**
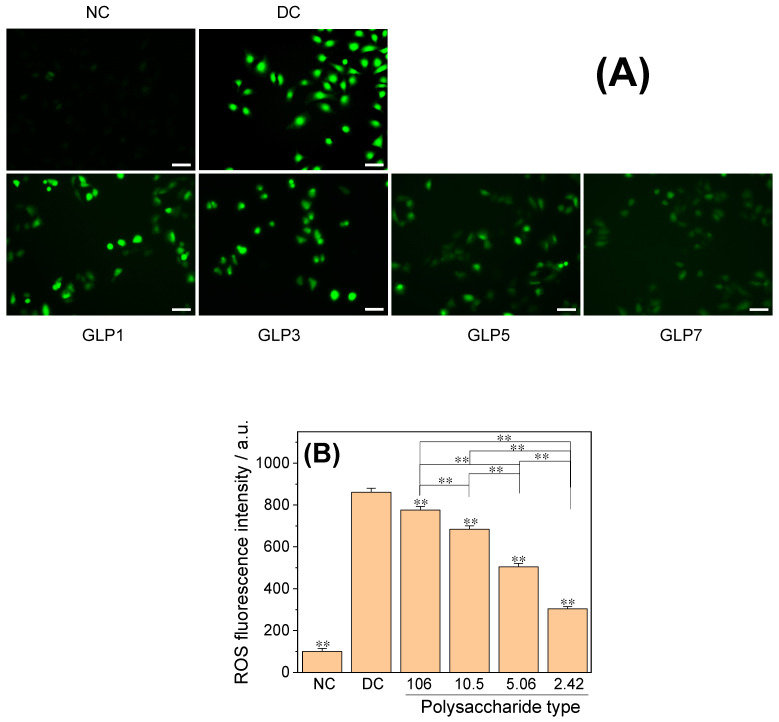
Effects of CaOx crystals regulated by GLPs at a concentration of 1.0 g/L on ROS level in HK-2 cells. (**A**) ROS fluorescence map; (**B**) fluorescence intensity quantitative diagram. The *M*_w_ of GLP1, GLP3, GLP5 and GLP7 was 106, 10.5, 5.06 and 2.42 kDa, respectively. NC: normal control group; DC: crystal control group without polysaccharides. Scale: 50 μm. Compared with DC, ** *p* < 0.01.

**Figure 10 foods-12-01031-f010:**
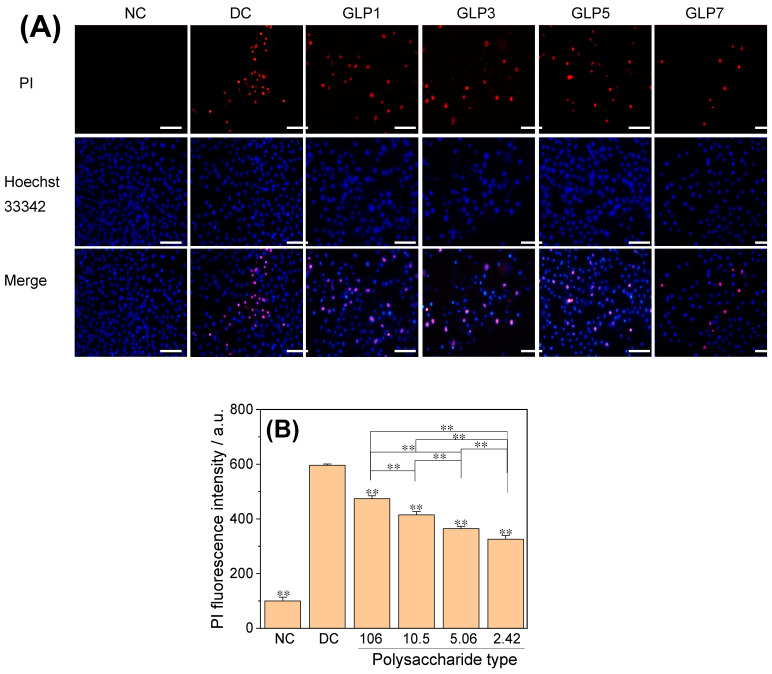
Effects of CaOx crystals regulated by GLPs at a concentration of 1.0 g/L on apoptosis and necrosis of HK-2 cells. (**A**) PI fluorescence map; (**B**) fluorescence intensity quantitative diagram. The *M*_w_ of GLP1, GLP3, GLP5 and GLP7 was 106, 10.5, 5.06 and 2.42 kDa respectively. NC: normal control group; DC: crystal control group without polysaccharides. Scale: 20 μm. Compared with DC, ** *p* < 0.01.

**Figure 11 foods-12-01031-f011:**
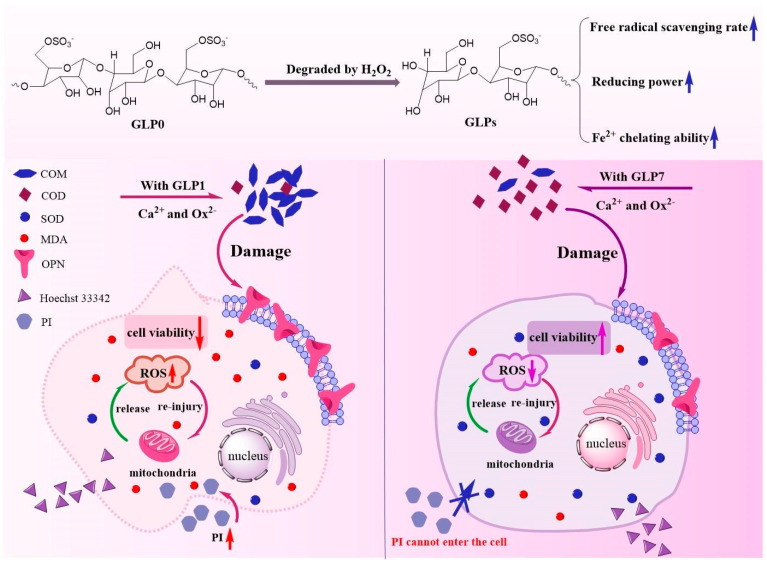
Model diagram of GLPs antioxidant reducing the cytotoxicity of CaOx crystals on HK-2 cells. The left side of the diagram is the toxic effect of CaOx regulated by high *M*_w_ GLP1 on the cells, where the formed COM crystals are more toxic and cause cell damage and cell contraction. The right side is the toxic effect of CaOx regulated by low *M*_w_ GLP7 on the cells, where the mainly formed COD crystals have less toxicity and less damage to the cells. The cell morphology is basically normal.

**Table 1 foods-12-01031-t001:** Seven GLPs fractions with different molecular weights and their –OSO_3_^–^ content.

Polysaccharide Abbreviation	GLP0	GLP1	GLP2	GLP3	GLP4	GLP5	GLP6	GLP7
Mean molecular weights *M*_w_/kDa	622	106	49.6	10.5	6.14	5.06	3.71	2.42
–OSO_3_^–^ content/%	13.07	13.37	13.41	13.46	13.55	13.55	13.56	13.46

**Table 2 foods-12-01031-t002:** Thermogravimetric analysis of CaOx crystals regulated by GLPs with concentration of 1.0 g/L.

Polysaccharide Abbreviation	Stage I/%	Stage II/% *^1^	Stage III/%	Stage IV/%	Residue/%
COM	12.33	0	19.18	30.13	38.36
DC	12.82	0	18.42	29.65	39.11
GLP1	13.68	5.04	13.59	29.47	38.22
GLP3	15.50	6.31	11.41	27.64	39.14
GLP5	17.26	8.81	10.44	27.15	36.34
GLP7	17.50	9.49	9.93	26.40	36.68

[*^1^] Stage II is the loss of polysaccharide, that is, the percentage of polysaccharide embedded in the crystals.

## Data Availability

All the data supporting the results were shown in the paper and can be applicable from the corresponding author.

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
