# Peer review of "Antioxidant Activities and Cytotoxicity of the Regulated Calcium Oxalate Crystals on HK-2 Cells of Polysaccharides from Gracilaria lemaneiformis with Different Molecular Weights"

_foods, 2023, doi:10.3390/foods12051031_

Round 1

Reviewer 1 Report

The manuscript shows the antioxidant activities of seven polysaccharides with different molecular weights from Gracilaria lemaneiformis.

1-     What the authors mean when refer to “repair effects” and “cytotoxicity difference of CaOx crystals induced by regulation”? (lines 79 and 84)

2-     Legends of Fig 5 and Fig 11 should contain more information.

Reviewer 2 Report

Dear Sir,

the overall impression after reviewing the paper that it contains possible questions and answers that can be asked after the research conducted. It is a bit strangely written just because of the questions that the authors themselves ask and answer. In order to write a scientific paper, it is not necessary to write the same. In other words, avoid asking questions and giving answers to them.

As for the actual arrangement of the chapters and subchapters, i.e., the hierarchy, you must choose a better way than numbering). [e.g., 1).] This is not a good way and try to find a better, i.e. common way of presenting subchapters.

Line 283: Do not start the sentence with ∙OH, but the hydroxyl radical.

Line 597: It is not a hydroxide group, but a hydroxyl one.

As for the polysaccharides themselves, which have different molecular weights, their composition is not apparent in the paper, except that they are basically β-1,3- and α-1,4-D and L-galactopyranose subunits. A more detailed composition is presented in the cited work. Indeed, the chemical composition of the polysaccharide itself may influence many of the properties investigated in these studies. The authors themselves talk about this in their paper, and it includes possible branching and side branches, as well as groups that can be attached to the carbohydrate part. I miss the connection between the structure and the interpretation of the results. There is some writing, but I think it could be better.

Sincerely,

Reviewer 3 Report

The authors have degraded polysaccharides extracted from Gracilaria lemaneiformis into various molecular weights and evaluated their respective antioxidant capacity and calcium oxalate crystal-induced cytotoxicity (HK-2 cell). The results showed that the lowest molecular weight, GLP7, had the lowest cytotoxicity for calcium oxalate crystal induction. This suggests that GLP7 may have potential as a preventive and therapeutic agent for renal calculi. Due to the large amount of data, conclusions are followed from multiple angles. However, there are a few points of concern.

Comments

Title

(1)     The English text of the title seems incorrect. How about the following? Antioxidant activities and cytotoxicity of the induced calcium oxalate crystals on HK-2 cells of polysaccharides from Gracilaria lemaneiformis with different molecular weights"

Materials and Methods

(2)     No statistical analysis was performed on the data obtained. I believe that statistical analysis is needed for many data.

Results

(3)     Fig.1: Alphabetize in order from the top of the diagram.

(4)     Fig.2: Is there any XRD data on the COM crystals? This result would help the reader to understand.

(5)     L370-389: It is difficult to tell which sentence describes which graph in Figure 2. Please make it easy to understand.

(6)     Fig.3: Can you tell from this image data alone that a COM has been formed? I doubt it; it seems to me that you need to show SEM images of the COM or cite other references.

(7)     Fig.4: No statistical processing has been performed. Please perform statistical processing.

(8)     Fig.6: No statistical processing has been performed. Please perform statistical processing.

(9)     Fig.7: No statistical processing has been performed. Please perform statistical processing.

(10)  L484: Cell viability is reduced in the DC group. Therefore, SOD activity may have decreased because the cells are dead; adding GLP does not increase SOD activity, but rather SOD activity is higher because cell viability is higher. If so, I think the wording needs to be changed.

(11)  L484: I'm curious about the phrase "MDA release"; MDA is not a substance that is released outside the cell, but rather accumulates inside the cell. Please revise the phrase.

(12)  Fig.8: Please quantify and graph the OPN coloration. Please also provide statistics on the results.

(13)  Fig.9: Please quantify and graph the ROS coloration. Please also provide statistics on the results.

(14)  Fig.8: Please quantify and graph the apoptosis and necrosis coloration. Please also provide statistics on the results.

Discussion

(15)  Table1: This is a result and should be explained in the Results section.

(16)  HK-2 test: Are there any COD results as well as COM results? The authors conclude that GLP changes COM to COD, which is less cytotoxic. However, there are no results in the authors' data that clarify whether COD is less cytotoxic.

(17)  The relationship between CaOx crystals and the antioxidant capacity of GLP is not explained. Please explain this point.

Reviewer 4 Report

This paper is continuation of former studies of the antioxidant activity difference of polysaccharides. It concerns he regulation effect of polysaccharides on the growth of CaOx crystals and the cytotoxicity difference of CaOx crystals induced by regulation. The main aim of this manuscript was to provide enlightenment for finding the best activity of polysaccharides for the prevention and treatment of kidney stones.

As kidney stone is a disease that seriously endangers people’s health this paper should be interesting for quite wide audience. But it would be nice if Authors could give the numbers not only for Chinese population vulnerable to this danger but also  the part of world population to show that this affects the human population.

The methods are decsribed in a very detailed way. Results presented properly - clearly, with easy to read figures and good despription, thougt the titles of figures are in too big font.

I wouls suggest to change the begning of the sentence in line No. 511: "Normal control group cell surface only weak green fluorescence; " it sounds like a mental shortcut and not the precise description.

I would suggest to put the tables 1 and 2 to the part of results. 

As MDPI does not give the precise information about selfcitation - "Authors should not engage in excessive self-citation of their own work." I did not marked YES with the part of "Did you detect inappropriate self-citations by authors?" though I was really tempted to do it. In literature (Self-citation in Publishing, DOI 10.1007/s11999-010-1480-8) one can find that  "Self-citation ranges from 7% to 20% of an article’s references. .... Thomson Reuters considers self-citation beyond 20% as suspect of abuse ...".

As Ouyang JM paper share is 23.7% of references it is up to Editor to decide if this is OK or too much and if this manuscript should be accepted or rejected.

Round 2

Reviewer 3 Report

I am satisfied with the revisions that have been made by the authors.